# Leveraging mHealth and Wearable Sensors to Manage Alcohol Use Disorders: A Systematic Literature Review

**DOI:** 10.3390/healthcare10091672

**Published:** 2022-09-01

**Authors:** Clemens Scott Kruse, Jose A. Betancourt, Stephanie Madrid, Christopher William Lindsey, Vanessa Wall

**Affiliations:** School of Health Administration, Texas State University, San Marcos, TX 78666, USA

**Keywords:** substance use disorder, alcohol use disorder, wearable sensors, mHealth, eHealth, telemedicine

## Abstract

Background: Alcohol use disorder (AUD) is a condition prevalent in many countries around the world, and the public burden of its treatment is close to $130 billion. mHealth offers several possible interventions to assist in the treatment of AUD. Objectives: To analyze the effectiveness of mHealth and wearable sensors to manage AUD from evidence published over the last 10 years. Methods: Following the Kruse Protocol and PRISMA 2020, four databases were queried (PubMed, CINAHL, Web of Science, and Science Direct) to identify studies with strong methodologies (*n* = 25). Results: Five interventions were identified, and 20/25 were effective at reducing alcohol consumption. Other interventions reported a decrease in depression and an increase in medication compliance. Primary barriers to the adoption of mHealth interventions are a requirement to train users, some are equally as effective as the traditional means of treatment, cost, and computer literacy. Conclusion: While not all mHealth interventions demonstrated statistically significant reduction in alcohol consumption, most are still clinically effective to treat AUD and provide a patient with their preference of a technologically inclined treatment Most interventions require training of users and some technology literacy, the barriers identified were very few compared with the litany of positive results.

## 1. Rationale

Alcohol use disorder (AUD) is characterized by the inability to stop or control alcohol use despite social, occupational, or health consequences [1]. Approximately 85.6% of people aged 18 years and older in the U.S. reported they drank alcohol, 69.5% reported they drank in the last year, and 54.9% reported they drank in the last month. In a survey of primary care providers in the European Union, AUDs were prevalent in 11.8% of the population, which is 1.6 times the population estimate [2]. AUD is specifically attributed to 735,153 deaths in 2019, but indirectly associated with 7,599,264 when alcohol-related deaths are taken into consideration [3]. In the U.S., AUD is associated with $120 billion per year in medical costs in the US, and $7.6 billion in the EU [3,4].

Telemedicine is defined by the World Health Organization (WHO) as healing at a distance through information and communication technologies (ICT) [5]. Telemedicine provides clinical support, overcomes geographical boundaries, involves ICT, and has a goal to improve health outcomes. Telemedicine comes in many forms, but wearable sensors can be connected to apps on mobile devices. When these wearable sensors provide clinical data to providers, this falls under the scope of mHealth.

Treatments for AUD can be both inpatient and outpatient, and they often must be tailored to the individual [1]. Wearable sensors have the ability to observe behavior and physiological constructs and combine them with location tracking. Tracking gait and sweat can provide feedback on abstinence and intoxication [6,7]. The geographic location can provide pre-programmed text messages to warn against danger areas (proximity to establishments that sell alcohol) [8].

In general, a systematic literature review is conducted to summarize recent science on a particular subject. A continuous growth of research combined with the rapid growth of technology compels scientists to systematically summarize available research and synthesize evidence. These products form the basis for funded research, and they can provide a foundation for modifying evidence based practice. As of the writing of this systematic review, 13 funded grant opportunities exist in the area of alcohol use disorder in the USA alone. Technology often serves as a fulcrum of change, and many mHealth solutions exist to help manage alcohol use disorder. A systematic literature review at the intersection of mHealth and the treatment of alcohol use disorder seemed timely. A systematic review in 2020 analyzed 32 articles over a 5-year period [9]. This study found half of the interventions reported improvements in at least one outcome (reduced cravings, or alcohol use). Only two of the interventions utilized wearable sensors. The remainder were feedback apps for craving management, coping assistance, and tailored feedback [9].

Another systematic review published in 2020 analyzed 22 articles over 10 years [10]. The study team found that most interventions resulted in a positive outcome (reduced depression, increased satisfaction, increased accessibility, increase quality of life, and decreased cost. Interventions included mobile health apps, eHealth (computer programs), telephone intervention, and 2-way video [10].

## 2. Objectives

The purpose of this review is to analyze the effectiveness of mHealth and wearable sensors to manage AUD, compared with the outcomes of the same conditions under traditional, face-to-face (in person) treatment, from evidence published in peer-reviewed and indexed journals over the last ten years. Effectiveness will be measured as improvements in AUD cravings, decrease in alcohol consumption, and a positive rating in patient satisfaction.

## 3. Methods

### 3.1. Eligibility Criteria

Articles for analysis were published in the last 10 years in peer-reviewed academic journals, and published in the English language. They must include participants who are adults (18 years of age or older). Preferred methods were true experiments (RCT, etc.), but quasi-experimental, non-experimental, and qualitative studies were also accepted. Other systematic reviews were not accepted so as not to confound the results. Works that did not mention wearable sensors or mHealth to treat AUD were excluded. Studies with participants under age 18 were excluded. Studies that did not report results were excluded.

### 3.2. Information Sources

Four data sources were queried: PubMed (MEDLINE), Cumulative Index of Nursing and Allied Health Literature (CINAHL), Web of Science, and Science Direct and a focused journal search in the Journal of Addictive Medicine. These databases were chosen because they are well known, exhaustive, and easily accessible by those who want to duplicate the research. MEDLINE was excluded from all searches except PubMed. Searches were conducted on 8 January 2022.

### 3.3. Search Strategy

Our study team used the Medical Subject Heading (MeSH) feature of the National Library of Medicine to create a Boolean search string that combined key index terms: (mhealth OR telemedicine OR “mobile app” OR biosensors) AND (“alcohol use disorder” OR “AUD”). We used the same search string in all databases and the focused journal search. As close as databases would allow, we used the same filter strategies.

### 3.4. Selection Process

We used the Boolean search string in all databases, filtered the results, and screened the abstracts for applicability, in accordance with the Kruse Protocol [11]. The Kruse Protocol defines a systematic methodology to conduct an exhaustive summary of evidence and report in accordance with the PRISMA standard. Studies were removed that did not address the objective statement.

### 3.5. Data Collection Process

We used a standardized Microsoft Excel spreadsheet as a data extraction tool collecting additional data fields at each step. The Kruse Protocol standardized the spreadsheet. We used a series of three consensus meetings to confirm the group of studies for analysis, conduct the thematic analysis, and perform additional analysis [12]. Abstracts were screened and studies were analyzed by at a minimum two reviewers.

### 3.6. Data Items

The Kruse Protocol dictated we collect the following fields of data at each step: DB Source, Date of publication, author names, title, participants, experimental intervention, results, medical outcomes, study design, sample size, bias within study, effect size (Cohen d), sensitivity, specificity, F1, country of origin, statistics uses, patient satisfaction, barriers to adoption, strength of evidence and quality of evidence.

### 3.7. Study Risk of Bias Assessment

Each reviewer noted observed bias and assessed the quality of each study using the Johns Hopkins Nursing Evidence Based Practice tool (JHNEBP) [13]. This was done because bias can limit the external validity of studies [14].

### 3.8. Effect Measures

This study included both qualitative and quantitative studies. Due to the fact that we accepted this range of methodology, we were unable to standardize summary measures, as would be performed in a meta-analysis. Measures of effect are summarized in tables for those studies in which it was reported.

### 3.9. Synthesis Methods

This subheading is for meta-analyses—NOT for systematic reviews. It will be removed by the editor prior to publication.

### 3.10. Reporting Bias Assessment

The overall ratings of strength and quality from the JHNEBP tool provided an assessment of the applicability of the cumulative evidence. Observations of bias were discussed for their implications on their reported results.

### 3.11. Additional Analyses and Certainty Assessment

We performed a narrative or thematic analysis of the observations to convert observations into themes (an observation that occurred multiple times became a theme) [12]. We calculated a frequency of occurrence and report this in an affinity matrix. Reporting the frequency provided confidence in the data analyzed.

## 4. Results

### 4.1. Study Selection

Figure 1 illustrates the study selection process from the four databases and one targeted journal search. Using established methods, we calculated a kappa statistic (k = 0.96, almost perfect agreement) [15,16]. Figure 1 illustrates the initial search results of 786 and how we filtered and screened these down to the group for analysis (*n* = 25).

### 4.2. Study Characteristics

Following PRISMA 2020 guidance, we created a PICOS table to tabulate the participants, intervention, results, medical outcomes, and study design for each study analyzed. In the 25 studies analyzed, all used adults as participants, and the experimental intervention was some form of mHealth. Of the 25 studies, 14/25 (56%) used an mHealth app, 4/25 (16%) used telephone or interactive voice response, 3/25 (12%) used mHealth serious games or cognitive training delivered on mobile devices, and four studies used either mHealth SMS or mHealth + telephone (2/26 each, 8%). Of the 25 studies analyzed over a 10-year period, 2 were from 2012 [17,18], 1 was from 2013 [19], 2 were from 2014 [20,21], 1 was from 2015 [22], 3 were from 2016 [23,24,25], 4 were from 2017 [26,27,28,29], 2 were from 2019 [30,31], and 2020 [32,33], 4 were from 2021 [34,35,36,37], and 2022 [38,39,40,41]. Zero studies were from 2018. A graphical display of this evolution of studies is illustrated in Figure 2.

Table 1 summarizes the study characteristics. From the 25 studies analyzed, 14/25 (56%) were randomized controlled trials, 4/26 (16%) were true experiments, 2/26 (8%) were non-experimental and 2 were mixed-methods, and 2 were observational, and 1/26 (4%) was qualitative. Results showed a reduction in consumption in 15/31 (45%) results themes, but also no significant difference in treatment outcomes in 5/31 (16%) results themes. For multiple interventions the no-difference variable brings into question whether organizations should expend the energy and expense to train users and implement the intervention.

### 4.3. Risk of Bias within and across Studies

Because of the high number of RCTs and true experiments in the group of articles analyzed, the JHNEBP quality assessment tool identified 18/25 (72%) as Strength of Evidence I. Only 7/25 (28%) were classified as Strength of Evidence III. Similarly, the strong methodology, large sample sizes, and consistency of results caused the JHNEBP tool to identify 23/25 (92%) as Quality of Evidence A. Only 2/26 (8%) were classified as Quality of Evidence B.

Reviewers also made note of internal and external bias in the studies. All articles were conducted in either one or multiple regions of only one country, which is an indication of selection bias. This threatens the internal validity of the study. Furthermore, 10/25 (40%) observations of sample bias were identified because the sample used a disproportionate percentage of one gender or race. This form of bias threatens the external validity of the results.

### 4.4. Results of Individual Studies

Following the Kruse Protocol, reviewers recorded independent observations during data extraction. These observations were discussed in Consensus Meeting number two. Through the discussion of observations, a thematic analysis was performed to make sense of the data [12]. Reviewers identified themes and performed a second data extraction to ensure no themes were omitted. Table 2 tabulates the themes identified in the literature. Appendix A and Appendix B provide an observation-to-theme match. While there is some overlap between Results, Medical Outcomes, and Effectiveness, reviewers felt it was necessary to report them separately in order to highlight both similarities and differences between the studies. Appendix C provides the other observations made by reviewers (sample size, bias, effect size, country of origin, statistics used, and the JHNEBP observations of strength and quality of evidence).

### 4.5. Results of Syntheses

This subheading is for a meta-analysis, not for a systematic review. This section will be removed by the editor before publishing.

### 4.6. Additional Analysis and Certainty of Evidence

Affinity matrices were created to summarize the frequency and probability of occurrence of each theme or observations. Frequency and probability do not imply importance: They only state the probability the theme or observation would be identified in the group for analysis. As part of the thematic analysis, observations that occurred more than once were identified as themes. All others are listed as individual observations.

#### 4.6.1. Patient Satisfaction

Patient satisfaction was very positive for all studies. The reason for this may have been because participants had already presented themselves for treatment for AUD, therefore, they would be positively disposed toward most interventions. The exact modality may not have negatively affected patient satisfaction. This is a significant error of both internal and external validity, and this variable should not be used to form any conclusions about interventions.

#### 4.6.2. Results of Studies

Table 3 summarizes the results of studies compared with a control group. Table 2 identifies which studies did not have a control group. Five themes and three individual observations were identified by the reviewers for a total of 68 occurrences in the literature. Reduction in consumption was identified in 15/31 (48%) of the occurrences [18,23,26,27,30,32,33,34,35,36,37,38,39,40,41]. In 10/31 (32%) of the occurrences, the reduction was statistically significant, but in 5/31 (15%) of the occurrences, it was not statistically significant [21,22,29,31,33]. Three of 25 (10%) occurrences mentioned the intervention caused positive retention in treatment programs [17,24,28]. Furthermore, in 3/25 occurrences, the participants increased self-efficacy and scored better on the AUDIT [23,30,32]. In 2/31 (6%) occurrences, the intervention decreased binge drinking [26,27]. The following are individual observations that could not fit into a theme. One intervention used a Bayesian Network Model to predict relapses. This enabled providers to intervene through text, email, or phone. One intervention highlighted a high rate of acceptance among participants, which may have been related to the fact that participants already volunteered for treatment—the modality may not have played a significant part. One intervention noted positive frontal lobe function which could lead to a decrease in addiction behaviors [19,20,25].

##### Medical Outcome Commensurate with the Use of mHealth

Table 4 summarizes the medical outcomes observed. Ten themes and two individual observations were recorded commensurate with the adoption of (intervention) for a total of 34 occurrences. Many of these themes were like those highlighted in results. Only differences from results will be reported. Three interventions identified an increase motivation to change behavior as a result of the intervention. This occurred in 3/34 (9%) observations [28,34,38]. A high number of observations were unable to be fit into themes. One article mentioned a reduction in craving for alcohol. One mentioned an improved rate of depression indicators. One mentioned an improvement in dependence on alcohol. One highlighted an increase in medication compliance [17,18,19,33].

##### Effectiveness Themes and Observations

Table 5 summarizes the effectiveness themes and observations. Eight themes and six individual observations were recorded by reviewers for a total of 50 occurrences. Many of these themes overlapped with study results and medical outcomes. Only the differences will be reported. In four of the interventions, it was highlighted that these are equally as effective at treating AUD, so the decision to choose one method over the other could fulfil a patient’s preference, and this preference may increase the success of the intervention [18,21,22,33]. Two interventions were highlighted as low cost [17,18]. Two interventions resulted in sustained abstinence from drinking [34,35]. One intervention was noted as exceptionally good at providing education about AUD and healthy habits [24].

##### Barriers to the Adoption of mHealth and Wearable Sensors to Manage AUD

Table 6 summarizes the barriers to the adoption of mHealth and wearable sensors to manage AUD. Four themes and two observations were reported for a total of 34 occurrences. Almost every intervention would require additional teaching of users and provider teams to work with it. This theme occurred 22/34 (65%) occurrences [17,18,20,23,24,25,27,28,29,30,31,32,33,34,35,36,37,38,39,40,41]. In 4/34 (12%) occurrences, it was highlighted that the intervention was equally as effective at managing AUD as the traditional treatment methods, so change may not be necessary [21,22,29,31]. In four of the interventions, it was mentioned that cost could be a consideration in implementing it [22,29]. There were 2/34 (6%) occurrences of computer literacy or needed access to the Internet [19,41]. Finally, the two interventions that could not be fitted into themes: Must sustain the intervention for a long period to get positive results, and the intervention impacts the provider’s workload [34,41].

##### Interactions between Observations

Overall, mHealth apps mostly resulted in a reduction in alcohol consumption or a reduction in cravings [17,18,19,20,24,26,28,30,33,36,37,38,39,41]. The mHealth + telephone interventions had the same effect [35,40]. The mHealth SMS interventions had mixed results: They both reduced consumption of alcohol, but only one was a statistically significant decrease [22,23]. The telephone interactive voice intervention also showed mixed results: They all decreased alcohol consumption, but not all were statistically significant [21,29,32,34]. Finally, the mHealth with serious games or cognitive training showed the most promise with a younger population. This intervention also showed a decrease in alcohol consumption, and one of them highlighted an increase in frontal lobe function, which is theorized will decrease addiction [25,27,31].

## 5. Discussion

### 5.1. Summary of Evidence

Twenty-five studies published in the last 10 years were analyzed for implications to the adoption of mHealth and wearable sensors for the treatment of AUD. Five intervention themes were identified in the literature. Twenty of the 25 studies analyzed reported effectivness at reducing alcohol consumption or cravings [17,18,21,22,23,26,27,28,29,30,31,32,33,34,35,37,38,39,40,41], but some improvements were not statistically significant [21,22,29,31,33]. Using mHealth or wearable sensors, even if not statistically significant, can fulfil the preference of a patient, and this preference may increase the success of the intervention [18,21,22,33]. mHealth is effective at educating patients [24], is inexpensive [17,18], and can increase self-efficacy or self-determination of AUD patients [17,23,26,27,30,32,34,36,38,40,41]. Overall, mHealth offers a viable alternative to traditional treatments, and in some cases, the results are stronger than traditional care.

Practitioners should be comfortable adopting this intervention for the treatment of AUD. Although some training will be necessary for most mHealth interventions [17,18,20,23,24,25,27,28,29,30,31,32,33,34,35,36,37,38,39,40,41], the efficacy of the intervention is well supported by the literature. Providers should also be mindful that mHealth interventions could adversely affect their workload [41], and the intervention requires some computer literacy and access to the Internet or WiFi [19,41]. This intervention could be preferred by some patients and enabling their preference could positively affect outcomes.

Future research should explore why some of these interventions did not demonstrate a statistically significant reduction in alcohol consumption. There may have been customization of SMS messages or tailoring of the apps to cater to preferences of the patients. This may increase the efficacy of the intervention and decrease prevalence of AUD.

### 5.2. Limitations

Several reviewers were used in this study to control for confirmation bias. Unfortunately, a review is often limited to what can be found. To ensure studies were of high quality, we only accepted studies that had been published, however, this subjects the study to publication bias because we did not consider grey literature. The studies selected all exhibited small instances of selection and sample bias, which affect their internal and external validity, respectively. We only selected published articles from the last 10 years because technology advances so rapidly. Had we looked back 15 years, we may have identified additional themes in the literature.

## 6. Conclusions

mHealth and wearable sensors are effective tools to decrease alcohol consumption, increase self-efficacy and self-determination, and provide overall treatment of AUD. The evolution of studies on this topic has slowly grown over time. mHealth technology may require additional training of users at both ends, but its low cost and efficacy outweigh the disadvantages. Although some interventions are not statistically different from traditional care, the use of mHealth and wearable sensors may fulfill the preference of a patient and increase the success of treatment.

## Figures and Tables

**Figure 1 healthcare-10-01672-f001:**
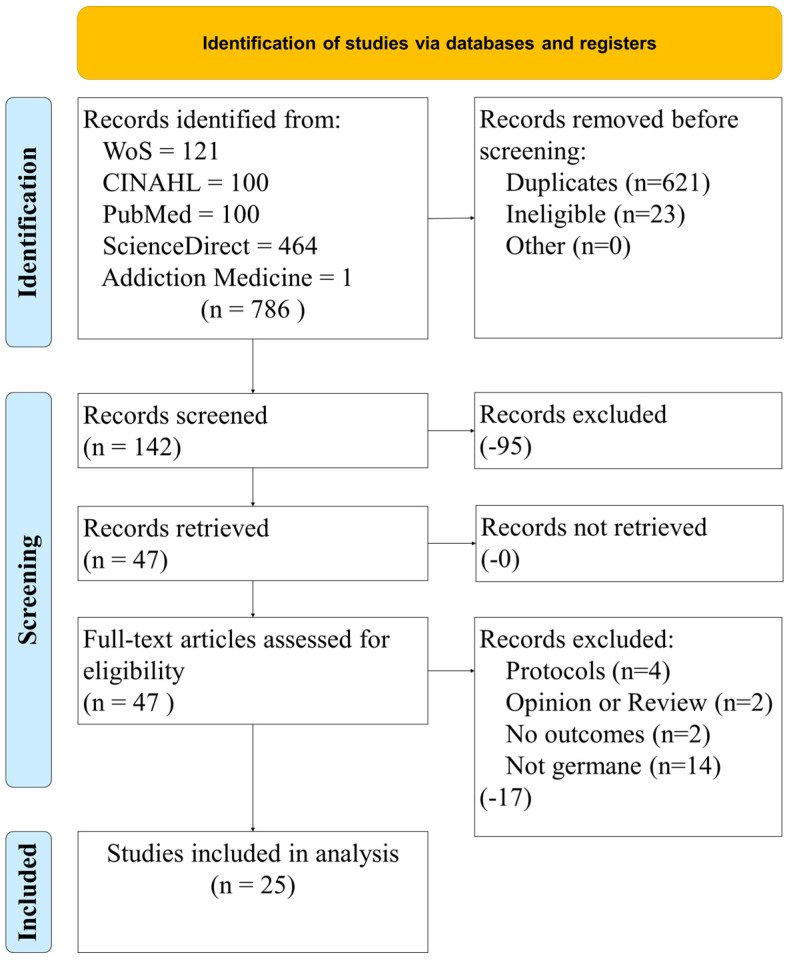
Study selection process.

**Figure 2 healthcare-10-01672-f002:**
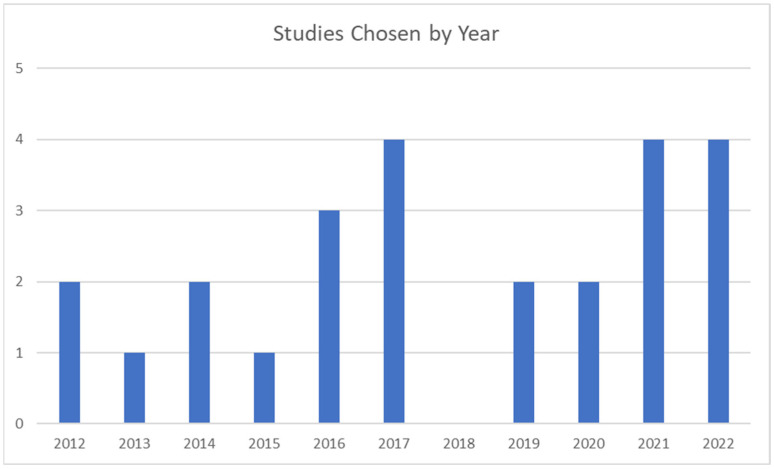
Evolution of studies chosen for analysis.

**Table 1 healthcare-10-01672-t001:** PICOS.

Authors	Participants	Experimental Intervention	Results	Medical Outcomes Reported	Study Design (See the List Below)
McTavish et al. [17]	Adults, average age 38.3, 60.6% male, 82.9% Caucasian	mHealth app to control SUD and AUD	94% used the app 1st week, and 80% used continued to use at week 16	Participants with AUD will use an app to manage their condition, App decreased cravings	True experiment
Murray et al. [18]	Adults ≥ 18, affluent area outside of London	mHealth app Down Your Drink (DYD)	No control group. Of those still using the app at 12 month, the reduction in drinking was 35 units	Reduction in consumption, reduction in AUD identification test, reduction in Leeds Dependence Questionnaire (LDQ, dependence), no significant change in Clinical Outcomes in Routine Evaluation (CORE-10, mental health) or EQ-5D (quality of life).	Mixed-Methods
Morgan et al. [19]	Adults ≥ 18	mHealth Internet-based app	Internet-based recruitment to mental health interventions is feasible	Improved rates of depression using intervention	RCT
Chih et al. [20]	Adults ≥ 18, average age 38, 62% male, 83% Caucasian	mHealth (A-CHESS) and BN	No control group. Responses to weekly check-in on A-CHESS can be a predictor of relapse	The prediction of lapse in sobriety gives counselors the chance to intervene through text, email, or phone call	Qualitative
Kalapatapu et al. [21]	Adults ≥ 18, average age 43.7, 87% female	telephone	Face-to-face cognitive based therapy (CBT) and T-CBT groups were similar on all treatment adherence outcomes and depression outcomes at all time points	telehealth means of treating is equally as effective as traditional therapy	True experiment
Stoner et al. [22]	Adults ≥ 18 (22–55), average age 37.5, 34.5% female	mHealth SMS	Adequate adherence ≥80% at week 8, not statistically significant between groups	SMS messages do not improve medication adherence, but equally as effective as traditional treatment to reduce consumption	RCT
Bock et al. [23]	Adults ≥ 18, average age 22, 61.3% female	mHealth SMS, text-message alcohol program (TMAP)	At week 6–12, TMAP participants less likely to report heavy drinking and negative alcohol consequences. Increased self-efficacy to resist drinking.	SMS effective at reducing consumption and increasing self-efficacy	True experiment
Freyer-Adam et al. [24]	Adults ≥ 18 (18–64)	mHealth Internet-based app	reached individuals and helped retain them in AUD programs	Not reported	RCT
Gamito et al. [25]	Adults ≥ 18, average age 45.45, 90% male	mHealth cognitive stimulation program (CSP) serious games	Cognitive ability between groups not statistically significant, but frontal-lobe function (Frontal Lobe Assessment, FAB) was significantly improved in the intervention group	Improvement in FAB with mHealth intervention	RCT
Barrio et al. [26]	Adults ≥ 18, average age 48, 50% female	mHealth app, SIDEAL	Reduced binge drinking and mean daily consumption, participants achieved their self-imposed objectives	Significant reduction in alcohol consumption	Non-experimental (no randomization, no control)
Gajecki et al. [27]	Adults ≥ 18, students, 66.7% female	mHealth app (skills training)	Reduced binge drinking and mean daily consumption, participants achieved their self-imposed objectives	Reduced alcohol consumption	RCT
Glass et al. [28]	Adults ≥ 18	mHealth app (A-CHESS)	Intervention showed increased odds of outpatient addiction treatment across follow-ups, but not mutual help	Reduced alcohol consumption, increased treatment	RCT
Rose et al. [29]	Adults ≥ 18	interactive voice response (IVR) brief intervention (BI)	Reduced alcohol consumption, but not statistically different than control	Reduced alcohol consumption	RCT
Jo et al. [30]	Adults ≥ 18	mHealth (online-based brief empowerment for alcohol-use monitor, on-BEAM)	Intervention group reported consuming less alcohol during the past week and lower AUDIT score	Reduced alcohol consumption	RCT
Mellentin et al. [31]	Adults ≥ 18	mHealth (cue exposure)	No differences were detected between the two experimental CET groups on any outcomes	Reduced alcohol consumption	RCT
Harder et al. [32]	Adults ≥ 18	mHealth (motivational interviewing)	Average AUDIT scores were lower for the intervention group	Reduced alcohol consumption, increased self-efficacy	RCT
Hendershot et al. [33]	Adults ≥ 18 (21–55)	mHealth feedback, opioid receptor gene (OPRM1)	OPRM1 genotype moderated the association of daily adherence with reduced same-day consumption (*p* = 0.007) and craving (*p* = 0.06), with these associations being stronger for participants with the 118 G variant. OPRM1 genotype did not moderate changes in craving and consumption over time	high-density assessments and person-centered analytic approaches, including modeling within-person variation in medication adherence, could be advantageous for pharmacogenetic studies	Non-experimental (no randomization, no control)
Constant et al. [34]	Adults ≥ 18	telephone	Study group had better alcohol abstinence rates than control	Intervention improves patient coping skills and motivation to modify alcohol use behaviors	RCT
Graser et al. [35]	Adults ≥ 18 (69% male)	telephone and smartphone-based intervention	Telephone-based intervention was more effective than text-based intervention	Sustained abstinence from excessive drinking occurred in telephone intervention group	RCT
Hammond et al. [36]	Adults ≥ 18 (61% male)	mHealth app	Intervention group utilized mobile app more effectively than control group	Complemented community substance use intervention programs	True Experiment
Manning et al. [37]	Adults ≥ 18 (58% female)	mHealth app	Intervention group reduced alcohol consumption rates	Improved alcohol consumption rates	Observational
Howe et al. [38]	Adults ≥ 18 (85% female; 62% Caucasian)	mHealth app	Use of mobile app improved decision making of study group participants	Mobile data collection can positively influence drinking decisions	Observational
Leightley et al. [39]	Adults > 18 (95% male; 100% Veterans)	mHealth app	Use of mobile app reduced rate of alcohol consumption among Veterans in study group	Reduced rates of alcohol consumption	RCT
McKay et al. [40]	Adults ≥ 18 (71% male; 82% African American)	telephone and smartphone-based intervention	Use of telephone or smartphone was effective in treating AUD	Improved rates of alcohol dependent persons	RCT
O’Grady et al. [41]	Adults ≥ 18 (Quant = 87% male/13% female; Qual = 43% male/57% female)	mHealth app	Provider-facing technology is effective alcohol intervention services and increased access to care in low- and middle-income countries.	Improved rates in alcohol dependent persons	Mixed Methods

**Table 2 healthcare-10-01672-t002:** Summary of analysis, sorted most chronologically.

Authors	Intervention Theme	Results Themes	Medical Outcomes Themes	Effectiveness Themes	Barrier Themes
McTavish et al. [17]	mHealth app	Good retention	Reduction in cravings	Low cost	Must train users
				Good acceptance	
				Decreased consumption/cravings	
				Increased self-efficacy/self-determination	
Murray et al. [18]	mHealth app	Reduction in consumption	Reduction in consumption	Low cost	Cost
			Improvement in dependence	Decreased consumption/cravings	Must train users
			No significant difference in treatment outcomes	Equally as effective as traditional care (preference)	
				Good acceptance	
Morgan et al. [19]	mHealth app	High rates of acceptance	Improved rates of depression	Decreased depression symptoms	Computer literacy/access to Internet
					Cost
Chih et al. [20]	mHealth app	Computer models can predict relapse	With prediction of relapse, providers can intervene	Can predict relapse and enable intervention	Must train users
Kalapatapu et al. [21]	Telephone/Interactive voice response	No significant difference in treatment outcomes	No significant difference in treatment outcomes	Decreased consumption/cravings	Equally as effective, so change may not be necessary
				Equally as effective as traditional care (preference)	
Stoner et al. [22]	mHealth SMS	No significant difference in treatment outcomes	No significant difference in treatment outcomes	Decreased consumption/cravings	Equally as effective, so change may not be necessary
				Equally as effective as traditional care (preference)	Cost
Bock et al. [23]	mHealth SMS	Reduction in consumption	Reduction in consumption	Decreased consumption/cravings	Must train users
		Increased self-efficacy	Increased self-efficacy	Increased self-efficacy/self-determination	
Freyer-Adam et al. [24]	mHealth app	Good retention	Not reported	Educates	Must train users
				Increased retention in treatment program	
Gamito et al. [25]	mHealth serious games/cognitive training	Positive frontal lobe function (FAB)	Increased frontal lobe function	Increased frontal lobe function	Must train users
Barrio et al. [26]	mHealth app	Decreased binge drinking	Reduction in consumption	Decreased consumption/cravings	Must train users
		Reduction in consumption		Increased self-efficacy/self-determination	
Gajecki et al. [27]	mHealth serious games/cognitive training	Decreased binge drinking	Reduction in consumption	Decreased consumption/cravings	Must train users
		Reduction in consumption		Increased self-efficacy/self-determination	
Glass et al. [28]	mHealth app	Good retention	Reduction in consumption	Decreased consumption/cravings	Must train users
			Increased motivation to change	Increased retention in treatment program	
Rose et al. [29]	Telephone/Interactive voice response	No significant difference in treatment outcomes	Reduction in consumption	Decreased consumption/cravings	Cost
					Equally as effective, so change may not be necessary
					Must train users
Jo et al. [30]	mHealth app	Reduction in consumption	Reduction in consumption	Decreased consumption/cravings	Must train users
		Increased self-efficacy	Increased self-efficacy	Increased self-efficacy/self-determination	
Mellentin et al. [31]	mHealth serious games/cognitive training	No significant difference in treatment outcomes	Reduction in consumption	Decreased consumption/cravings	Equally as effective, so change may not be necessary
					Must train users
Harder et al. [32]	Telephone/Interactive voice response	Reduction in consumption	Reduction in consumption	Decreased consumption/cravings	Must train users
		Increased self-efficacy	Increased self-efficacy	Increased self-efficacy/self-determination	
Hendershot et al. [33]	mHealth app	Reduction in consumption	Reduction in consumption	Increased medication compliance	Must train users
		No significant difference in treatment outcomes	Increased medication compliance	Equally as effective as traditional care (preference)	
				Decreased consumption/cravings	
Constant et al. [34]	Telephone/Interactive voice response	Reduction in consumption	Reduction in consumption	Decreased consumption/cravings	Must train users
			Increased motivation to change	Increased self-efficacy/self-determination	Must sustain intervention for long-term results
				Sustained abstinence from drinking	
Graser et al. [35]	mHealth + telephone	Reduction in consumption	Reduction in consumption	Decreased consumption/cravings	Must train users
				Sustained abstinence from drinking	
Hammond et al. [36]	mHealth app	Reduction in consumption	Reduction in consumption	Increased self-efficacy/self-determination	Must train users
Manning et al. [37]	mHealth app	Reduction in consumption	Reduction in consumption	Decreased consumption/cravings	Must train users
Howe et al. [38]	mHealth app	Reduction in consumption	Reduction in consumption	Increased self-efficacy/self-determination	Must train users
			Increased motivation to change	Decreased consumption/cravings	
Leightley et al. [39]	mHealth app	Reduction in consumption	Reduction in consumption	Decreased consumption/cravings	Must train users
McKay et al. [40]	mHealth + telephone	Reduction in consumption	Reduction in consumption	Increased self-efficacy/self-determination	Must train users
				Decreased consumption/cravings	
O’Grady et al. [41]	mHealth app	Reduction in consumption	Reduction in consumption	Increased self-efficacy/self-determination	Computer literacy/access to Internet
				Decreased consumption/cravings	Impacts provider workload
				Increased access to care	Must train users

**Table 3 healthcare-10-01672-t003:** Results to the studies.

Results Themes and Observations	Frequency
Reduction in consumption [18,23,26,27,30,32,33,34,35,36,37,38,39,40,41]	15
No significant difference in treatment outcomes [21,22,29,31,33]	5
Good retention in treatment [17,24,28]	3
Increased self-efficacy [23,30,32]	3
Decreased binge drinking [26,27]	2
Computer models can predict relapse [20]	1
High rates of acceptance [19]	1
Positive frontal lobe function (FAB) [25]	1
	31

**Table 4 healthcare-10-01672-t004:** Medical outcomes commensurate with the adoption of mHealth.

Medical Outcomes Themes and Observations	Frequency
Reduction in consumption [18,23,26,27,28,29,30,31,32,33,34,35,36,38,39,40,41]	18
Increased self-efficacy [23,30,32]	3
Increased motivation to change [28,34,38]	3
No significant difference in treatment outcomes [18,21,22]	3
Reduction in cravings [17]	1
With prediction of relapse, providers can intervene	1
Improved rates of depression [19]	1
Improvement in dependence [18]	1
Increased frontal lobe function [25]	1
Increased medication compliance [33]	1
Not reported [24]	1
	34

**Table 5 healthcare-10-01672-t005:** Effectiveness Themes and Observations.

Effectiveness Themes and Observations.	Frequency
Decreased consumption/cravings [17,18,21,22,23,26,27,28,29,30,31,32,33,34,35,37,38,39,40,41]	20
Increased self-efficacy/self-determination [17,23,26,27,30,32,34,36,38,40,41]	11
Equally as effective as traditional care (preference) [18,21,22,33]	4
Increased retention in treatment program [17,24,28]	3
Low cost [17,18]	2
Good acceptance [17,18]	2
Sustained abstinence from drinking [34,35]	2
Increased frontal lobe function [25]	1
Increased access to care [41]	1
Decreased depression symptoms [19]	1
Increased medication compliance [33]	1
Educates [24]	1
Can predict relapse and enable intervention [20]	1
	50

**Table 6 healthcare-10-01672-t006:** Barrier themes and observations.

Barriers Themes and Observations	Frequency
Must train users [17,18,20,23,24,25,27,28,29,30,31,32,33,34,35,36,37,38,39,40,41]	22
Equally as effective, so change may not be necessary [21,22,29,31]	4
Cost [22,29]	4
Computer literacy/access to Internet [19,41]	2
Must sustain intervention for long-term results [34]	1
Impacts provider workload [41]	1
Not reported	0
	34

## Data Availability

Data from this study can be obtained by asking the lead author.

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
