# Peer review of "Leveraging mHealth and Wearable Sensors to Manage Alcohol Use Disorders: A Systematic Literature Review"

_healthcare, 2022, doi:10.3390/healthcare10091672_

Round 1

Reviewer 1 Report

I find the subject of the study, focusing on health prevention measures, very interesting. In any case, I would like to point out a few comments after reviewing the paper that I find interesting.

TITLE. Well defined, attractive and related to what the research is about.

ABSTRACT. Although I understand that in a summary the authors cannot go into detail, I would add some explanation. For example, when the researchers refer in line 12 to a "global problem", what do they mean, what kind of problem, a problem that affects all geographical areas? KEYWORDS. In the case of "substance use disorder", fine, although I miss "alcohol use disorders", a term that would be more refined in the key concepts.

——————————————

The rationale, although well structured, seems to me interesting enough to be expanded and to justify why the authors consider it necessary to carry out a systematic review. I think the argument of line 44 is relevant and the tailoring of treatments to the individual could be completed with the motivational aspect.

I have doubts about the way in which the researchers have bounded the study geographically, as I do not understand why in some paragraphs they refer to the U.S., in others to the European Union and in others they speak in a globalised way.

What are the bibliographical references of the systematic reviews published in 2020 to which the authors refer in lines 50 and 55? If they are [9] and [10], I think they should be included in the lines referred to.

As far as the objectives are concerned, I think that increasing the quality of life is a very subjective concept. I believe it would be better defined if it could be specified or clarified what the authors are referring to in line 65.

In addition, it would explain what exactly is meant by "evidence published" in line 63.

On methodology (as in line 63), I would specify in line 68 whether the evidence published refers to publications in scientific journals or in a specific category, indexed or not, .....

Can other true experiments be added in addition to RCTs (line 70)?

What are the reasons for choosing the four data sources indicated in the study?

Although the bibliographic reference to the Kruse Protocol is included in line 88, it would be helpful to explain this succinctly in the text.

In line 107 it is not clear to me whether the authors want to refer to quantitative and qualitative studies.

In section 4.2 on results, a graph with the evolution of studies per year could be added. In addition to visually representing this aspect, it could be developed in the conclusions.

I do not know if it is the most appropriate thing to include so many fractions, when simply indicating the percentage of the sample in each case would simplify the writing.

Sometimes, as in the discussion section, the authors use somewhat vague terms such as in line 260 "almost all".

Probably this section is the one I find most interesting, as it sets out the scenario of a possible continuity of the study and future lines of work.

Section 6 seems to me to be short, as interesting conclusions could be added to the study.

Author Response

Reviewer 1
TITLE. Well defined, attractive and related to what the research is about.

ABSTRACT. Although I understand that in a summary the authors cannot go into detail, I would add some explanation. For example, when the researchers refer in line 12 to a 
"global problem", what do they mean, what kind of problem, a problem that affects all geographical areas? KEYWORDS. In the case of "substance use disorder", fine, although I 
miss "alcohol use disorders", a term that would be more refined in the key concepts.
** We changed the phrase to "condition prevalent in many countries around the world." We added "alcohol use disorder" to the key terms.
——————————————

The rationale, although well structured, seems to me interesting enough to be expanded and to justify why the authors consider it necessary to carry out a systematic review. 
I think the argument of line 44 is relevant and the tailoring of treatments to the individual could be completed with the motivational aspect.
** We added a paragraph to justify conducting a systematic literature review on this topic.

I have doubts about the way in which the researchers have bounded the study geographically, as I do not understand why in some paragraphs they refer to the U.S., 
in others to the European Union and in others they speak in a globalised way.
** Unfortunately, there is not a body of research that spans multiple countries. This is largely due to regionalized funding for research. When searching for research, we found only geographically bound research. If there is a body of research that provides global evidence, we would be happy to include it.

What are the bibliographical references of the systematic reviews published in 2020 to which the authors refer in lines 50 and 55? If they are [9] and [10], I think they should be included in the lines referred to.
** In addition to these references being included at the end of these paragraphs, we included them at the beginning.

As far as the objectives are concerned, I think that increasing the quality of life is a very subjective concept. I believe it would be better defined if it could be specified or clarified what the authors are referring to in line 65.
** We agree this is a nebulous term. It has been removed.

In addition, it would explain what exactly is meant by "evidence published" in line 63.
** We added "in peer reviewed and indexed journals" to clarify this phrase.

On methodology (as in line 63), I would specify in line 68 whether the evidence published refers to publications in scientific journals or in a specific category, 
indexed or not, .....
** We added "in peer reviewed and indexed journals" to clarify this phrase.

Can other true experiments be added in addition to RCTs (line 70)?
** This strong methodology is preferred because of the quality-assessment tool we used (Johns Hopkins Nursing Evidence Based Practice). The highest quality of evidence is the true experiment or an RCT.

What are the reasons for choosing the four data sources indicated in the study?
** We added, "These four databases are well known, exhaustive, and easily accessible by those who want to duplicate the research."

Although the bibliographic reference to the Kruse Protocol is included in line 88, it would be helpful to explain this succinctly in the text.
** We added, "The Kruse Protocol defines a systematic methodology to conduct an exhaustive summary of evidence and report in accordance with the PRISMA standard."

In line 107 it is not clear to me whether the authors want to refer to quantitative and qualitative studies.
** We modified this paragraph: This study included both qualitative and quantitative studies. Due to the fact that we accepted this range of methodology, we were unable to standardize summary measures . . ."

In section 4.2 on results, a graph with the evolution of studies per year could be added. In addition to visually representing this aspect, it could be developed in the conclusions.
** We added a graph (figure 2) to depict this evolution of studies, and we added a sentence in the conclusion about it.

I do not know if it is the most appropriate thing to include so many fractions, when simply indicating the percentage of the sample in each case would simplify the writing.
** As I understand it, the fractions are required by the editor.

Sometimes, as in the discussion section, the authors use somewhat vague terms such as in line 260 "almost all".
** We corrected this to twenty of twenty-five articles analyzed.

Probably this section is the one I find most interesting, as it sets out the scenario of a possible continuity of the study and future lines of work.

Section 6 seems to me to be short, as interesting conclusions could be added to the study.
** We added more to the conclusions.

Reviewer 2 Report

Dear Authors,

I have read the manuscript entitled “Leveraging mHealth and Wearable Sensors to Manage Alcohol Use Disorders: A Systematic Literature Review”. The review paper focuses on analyzing the effectiveness of mHealth and wearable sensors in the management of patients with alcohol use disorders. Therapeutic approaches based on mobile health as a medical practice use mobile devices such as mobile phones, patient monitoring devices as well as a range of wireless devices as tools.

The manuscript is valuable because the analysis shows that almost all studies were effective in reducing alcohol consumption.

The paper complies with the requirements of the journal, being based on a thorough and rigorous analysis, very well structured through the large number of tables. The bibliographic references used for documentation are in agreement with the chosen topic, the studies being from the last 10 years. Although they were not mandatory, I appreciate that you also gave us some pertinent conclusions.

I have a few questions though:

1. Some studies state that the accuracy of wearable sensors is 75-77%. This aspect could represent an inconvenience in obtaining true real data on patient monitoring. What do the authors think about this aspect?

2. What happens to patients after stopping monitoring through this type of application? Does alcohol relapse occur?

3. A major problem for patients with alcohol use disorders is stigmatization. Can this desideratum be resolved? If so, could the authors describe how?

4. Is the compliance of patients to give up alcohol consumption through such a technology more effective than through classical methods?

Author Response

Reviewer 2
I have read the manuscript entitled “Leveraging mHealth and Wearable Sensors to Manage Alcohol Use Disorders: A Systematic Literature Review”. The review paper focuses on analyzing the effectiveness of mHealth and wearable sensors in the management of patients with alcohol use disorders. Therapeutic approaches based on mobile health as a medical practice use mobile devices such as mobile phones, patient monitoring devices as well as a range of wireless devices as tools.

The manuscript is valuable because the analysis shows that almost all studies were effective in reducing alcohol consumption.

The paper complies with the requirements of the journal, being based on a thorough and rigorous analysis, very well structured through the large number of tables. The bibliographic references used for documentation are in agreement with the chosen topic, the studies being from the last 10 years. Although they were not mandatory, I appreciate that you also gave us some pertinent conclusions.

I have a few questions though:

1. Some studies state that the accuracy of wearable sensors is 75-77%. This aspect could represent an inconvenience in obtaining true real data on patient monitoring. What do the authors think about this aspect?
** Participants in the studies analyzed volunteered for treatment. Accuracy of sensors do not negate the desire to recover. 

2. What happens to patients after stopping monitoring through this type of application? Does alcohol relapse occur?
** The studies do not report behavior after the end of a study. Healthy behaviors are developed which help the participants resist the cravings.

3. A major problem for patients with alcohol use disorders is stigmatization. Can this desideratum be resolved? If so, could the authors describe how?
** mHealth provides relief in this area by enabling fewer in-office visits. This reduces the stigmatization. We covered this in a previous publication, 
Kruse CS, Lee K, Watson JB, Lobo LG, Stoppelmoor AG, Oyibo SE. Measures of effectiveness, efficiency, and quality of telemedicine in the management of alcohol abuse, addiction, and rehabilitation: systematic review. Journal of medical Internet research. 2020 Jan 31;22(1):e13252.

4. Is the compliance of patients to give up alcohol consumption through such a technology more effective than through classical methods?
** This is reported under the "Results" column, because this column is in comparison to a control group (or treatment as usual). In some cases, yes, it is more effective. However, not all studies analyzed included a control group (qualitative or observational studies). In these cases, a comparison was not provided.